# Understanding *Artemisia cina* Ethyl Acetate Extract’s Anthelmintic Effect on *Haemonchus contortus* Eggs and L_3_ Larvae: The Synergism of Peruvin Binary Mixtures

**DOI:** 10.3390/pathogens13060509

**Published:** 2024-06-16

**Authors:** Luis David Arango-De-la Pava, Manasés González-Cortazar, Alejandro Zamilpa, Jorge Alfredo Cuéllar-Ordaz, Héctor Alejandro de la Cruz-Cruz, Rosa Isabel Higuera-Piedrahita, Raquel López-Arellano

**Affiliations:** 1Facultad de Estudios Superiores Cuautitlán, Universidad Nacional Autónoma de México, Cuautitlán 54714, Estado de México, Mexico; luis.david@cuautitlan.unam.mx (L.D.A.-D.-l.P.); jcuellar@unam.mx (J.A.C.-O.); delacruz@unam.mx (H.A.d.l.C.-C.); 2Centro de Investigación Biomédica del Sur, Instituto Mexicano del Seguro Social, Xochitepec 62790, Morelos, Mexico; gmanases@hotmail.com (M.G.-C.); azamilpa_2000@yahoo.com.mx (A.Z.)

**Keywords:** Asteraceae, cinic acid, sesquiterpene lactone, pharmacodynamic interactions, synergism, binary mixtures

## Abstract

*Haemonchus contortus*, a blood-feeding parasite in grazing sheep, causes economic losses. Drug resistance necessitates exploring plant-based anthelmintics like *Artemisia cina* (Asteraceae). The plant, particularly its ethyl acetate extract, shows anthelmintic activity against *H. contortus*. However, there is limited information on pharmacodynamic interactions in ethyl acetate compounds. The study aims to identify pharmacodynamic interactions in the ethyl acetate extract of *A. cina* with anthelmintic effects on *H. contortus* eggs and L_3_ larvae using binary mixtures. Bioactive compounds were isolated via chromatography and identified using spectroscopic techniques. Pharmacodynamic interactions were assessed through binary mixtures with a main compound. Four bioactive compounds were identified: 1-nonacosanol, hentriacontane, peruvin, and cinic acid. Binary mixtures, with peruvin as the main compound, were performed. Peruvin/1-nonacosanol-hentriacontane and peruvin/cinic acid mixtures demonstrated 1.42-fold and 4.87-fold increased lethal effects in *H. contortus* L_3_ infective larvae, respectively, at a 0.50LC_25_/0.50LC_25_ concentration. In this work, we determined the synergism between bioactive compounds isolated from the ethyl acetate extract of *A. cina* and identified unreported compounds for the specie.

## 1. Introduction

*Haemonchus contortus* is recognized as one of the most pathogenic gastrointestinal nematodes, causing inflammation, mucosal secretions, and micro-hemorrhages in the abomasum of small ruminants. Its blood-feeding habits contribute significantly to global economic losses in flocks, leading to malnutrition, low feed conversion, anemia, reduced appetite, diminished fertility rates, and increased mortality in both young and mature animals [1]. Moreover, *H. contortus* can elude the host immune response, employing various genetic mechanisms to tolerate the toxicity of widely used anthelmintic drugs. This has resulted in the diminishing efficacy of anthelmintic drugs in many countries [2,3]. Consequently, the control of *H. contortus* has become a worldwide concern, prompting exploration into alternative control strategies beyond chemotherapy, such as the investigation of medicinal plants [4], such as *Artemisia cina*.

The efficacy of *A. cina* against *H. contortus* has been demonstrated both in vitro and in vivo. In vitro, the *n*-hexane extract (*n*-HE) of *A. cina* exhibited the highest larvicidal activity against transitional larvae L_3_–L_4_ of *H. contortus* [5]. In an in vivo study conducted on naturally infected periparturient goats, the administration of *n*-HE derived from *A. cina* resulted in a notable reduction in the fecal egg counts of *H. contortus* and *Teladorsagia circumcincta*. This extract was found to contain two previously unidentified compounds for *A. cina*, namely isoguaiacin and norisoguaiacin [6]. Recently, a bio-guided isolation of the ethyl acetate extract (EAE) was realized, allowing the identification of cinic acid, a new sesquiterpene with promising larvicidal activity against L_3_ of *H. contortus* infective larvae. [7]. However, a limitation of this bio-guided isolation is that it does not consider pharmacodynamic interactions, which are crucial when formulating pharmaceutical preparations.

Pharmacodynamics allows the study of the biochemical and physiological actions that arise from administering bioactive compounds, particularly about dose–response and their effects. When two bioactive molecules are administered together, they can affect different targets or the same targets, leading to additive, synergistic, or subadditive actions [8]. Additivity refers to the combined effects of each bioactive compound in a drug combination. Synergy occurs when the overall impact of the drug combination surpasses the anticipated outcome based on simple additivity. At the same time, subadditivity manifests when the observed effect of the drug combination is inferior to the cumulative impact of its components [9].

Plant extracts are a complex matrix of bioactive compounds, and it is not easy to understand their mechanism of action. The methodological approach involving binary mixtures with a main compound and compounds of interest leads to the discovery of pharmacological interactions desirable for formulating plant-based pharmaceutical preparations [10]. Therefore, this study aimed to determine the presence of pharmacodynamic interactions among bioactive compounds in the ethyl acetate extract of *A. cina* with anthelmintic effects on eggs and L_3_ infective larvae of *H. contortus*. This information will be utilized to establish the foundations for an anthelmintic pharmaceutical preparation based on *A. cina*.

## 2. Materials and Methods

### 2.1. Plant Material

The dried pre-flowering leaves and stems of *A. cina* O. Berg ex Poljakov (Asteraceae) (3.8 kg) were acquired from the Hunab laboratory. Dr. Alejandro Torres-Montúfar verified and authenticated a voucher specimen, which was then preserved in the herbarium of Facultad de Estudios Superiores Cuautitlán (FES-C) UNAM, México, assigned voucher number 11967. The plant was cultivated under controlled conditions of 80% humidity, a temperature of 24 °C, and soil with a pH of 6.3. The plant was cut and dried at room temperature for seven days at pre-flowering. It was cut into pieces of approximately 2 cm and packed in a plastic container until use.

### 2.2. Ethyl Acetate Artemisia cina Extract (EAE)

The extraction process was through maceration. A total of 1 kg of dried *A. cina* leaves and stems was finely ground and placed in a 1 L Erlenmeyer flask. The leaves and stems were extracted using 5 L of ethyl acetate (J.T. Baker^®^(Madrid, Spain), allowing the mixture to macerate for 72 h at room temperature (23–25 °C). The resulting extract was filtered through Whatman No. 4 paper, and the solvent was eliminated through low-pressure distillation using a rotary evaporator (DLAB RE-100 Pro, Beijing, China) operating at 40 °C and 100 rpm. The obtained extract was subsequently lyophilized (Biobase BK-FD10PT, Jinan, China ) and stored at four °C in refrigeration (Biobase BXC-V588M, Jinan, China) for further phytochemical and biological assays.

### 2.3. Isolation of Chemical Compounds from the Ethyl Acetate Extract

A total of 28.7 g of the EAE was separated using open-column chromatography. The stationary phase utilized was standard silica gel 60 (Merck, particle size: 0.015–0.040 mm), while a solvent gradient system consisting of *n*-hexane-ethyl acetate (J.T. Baker^®^) was employed. The column began with *n*-hexane and decreased concentrations of *n*-hexane and increased concentrations of ethyl acetate. A total of 40 samples was acquired and categorized into 11 fractions based on their chemical similarity, which was assessed through thin-layer chromatography. Subsequently, the samples were concentrated using a rotary evaporator. According to thin layer chromatography (TLC), fractions 3 (C1F3), 5 (C1F5), and 6 (C1F6) contained crystals that were washed with ethyl acetate (C1F3) and n-hexane (C1F5 and C1F6). C1F3, C1F5, and C1F6 crystals were analyzed by HPLC-DAD and GC-MS. C1F3 (81 mg, 0.28% yield) did not display UV spectra in HPLC-DAD, but in GC-MS, two prominent peaks represented 83.955% of the abundance of the sample (Appendix A). Compound **1** presented a retention time (RT) of 31.105 min (GC-MS), 31.343% of relative abundance (Appendix A), and [M + H]^+^ = 425 *m*/*z* (Appendix A); Compound **2** an RT of 33.938 min (GC-MS), 52.612% of relative abundance (Appendix A), and [M + H]^+^ = 437 *m*/*z* (Appendix A). C1F5 (Compound **3**) (87 mg, 0.30% yield) presented an RT of 24.786 min (Appendix A) and [M + H]^+^ = 283 *m*/*z* (Appendix A) (GC-MS), 15.748 min, and UV spectra λmax of 211.0 nm (Appendix A) (HPLC-DAD). C1F6 (Compound 4) (267 mg, 0.93% yield) 22.526 min and [M + H]^+^ = 265 *m*/*z* (Appendix A) (GC-MS), 14.208 min, and UV spectra λmax 216.9 nm (Appendix A) (HPLC-DAD).

### 2.4. TLC and HPLC Analyses

Analytical TLC was performed on precoated Merck silica gel 60F254 or RP-18F254 plates. Ceric sulfate reagent was used to visualize isolated compounds.

HPLC-DAD analyses used a Waters 2695 separations module with a Waters 2996 photodiode array detector. The HPLC analysis utilized a LiChrospher^®^ 100 RP-18 column (4 mm × 250 mm, five μm) from Merck, Kenilworth, NJ, USA. The mobile phase consisted of two solvent reservoirs: A (H_2_O-Trifluoroacetic acid 0.05%) and B (CH_3_CN). The gradient system employed was as follows: 0–8 min, transitioning from 100% to 0% B; 9–12 min, shifting from 90% to 10% B; 13–15 min, changing from 80% to 20% B; 16–20 min, varying from 70% to 30% B; 21–25 min, transitioning from 0% to 100% B; and 26–28 min, maintaining 100% B. The flow rate was set at 1 mL/min, and the injection volume was ten μL [10].

### 2.5. GC-MS Analysis

The GC-MS analysis utilized an Agilent Technologies HP 6890 gas chromatograph coupled with an MSD 5973 quadrupole mass detector (HP Agilent, Santa Clara, CA, USA). An HP-5MS capillary column was employed (30 m length, 0.25 mm inner diameter, and 0.25 µM film thickness). Helium served as the carrier gas at a constant 1 mL/min flow. The inlet temperature was maintained at 250 °C, and the oven temperature initially held at 40 °C for 1 min, then increased to 280 °C at ten °C/min intervals. The mass spectrometer operated in a positive electron impact mode (ionization energy: 70 eV) with selective ion monitoring. Identification and quantification were based on target ions, comparing mass spectra with the NIST library (version 1.7a). Relative percentages were determined by signal integration using GC Chem Station software (version C.00.01), with composition reported as a percentage of the total signal area [7].

### 2.6. NMR Experiments

Nuclear Magnetic Resonance (NMR) experiments such as ^1^H, COSY, HSQC, HMBC, and DEPTQ were realized using a Bruker AVANCE III HD at a frequency of 500 MHz. CD3COCD3, CDCl3, and C6D6 were used as solvents; tetramethylsilane (TMS) was employed as an internal standard. The chemical shifts (δ) are presented in ppm, and coupling constants are expressed in Hz.

### 2.7. In Vitro Assays with Haemonchus contortus Eggs and L_3_ Infective Larvae

Isolated compounds were then assessed for anthelmintic activity in lethal effect (LE) and inhibition of the larvae hatching through egg hatching inhibition (EHI) assays, conducted in vitro in a 96-well microplate with approximately 100 L_3_ larvae/eggs per well ° (n = 4), with three replicates under the same conditions. Two control groups were used: (a) distilled water and (b) ivermectin (5 mg/mL, Sigma Aldrich, Darmstadt, Germany). Peruvin and 1-nonacosanol/hentriacontane (21.3% and 52.6% of C1F3, respectively) were tested at five different concentrations (2.000, 1.000, 0.500, 0.250, and 0.125 mg/mL); cinic acid was tested at eight concentrations (2.000, 1.000, 0.500, 0.250, 0.120, 0.062, 0.003, and 0.001 mg/mL). The lethal effect in L_3_ infective larvae was evaluated 24 h post-exposure, and egg hatching inhibition 48 h post-exposure. Dead and alive larvae, the number of unhatched eggs (dead and larvated), and L_1_ larvae were counted to determine the mortality and egg hatching inhibition percentage [6]. A microscope with 10× magnification (Olympus^®^, model CK-2, Tokio, Japan) was used to count larvae and eggs.

### 2.8. Isobolograms of Binary Mixtures of Isolated Compounds from Ethyl Acetate Artemisia cina Extract

Isobolograms were performed to determine the type of pharmacodynamic interaction that presented the binary mixtures using a main compound (Mc). The Mc was chosen considering the hatching inhibition, lethal effect, yield percentage, and purity. The binary mixtures were prepared using subsequent dilutions of LC_25_ and evaluated in a fixed proportions scheme starting from the LC_25_ of the Mc and isolated compounds (Table 1) (1.00ALC_25_/0.00BLC_25_, 0.75ALC_25_/0.25BLC_25_, 0.50ALC_25_/0.50BLC_25_, 0.25ALC_25_/0.75BLC_25_, and 0.00ALC_25_/1.00BLC_25_), where A was the Mc and B was the other compounds isolated. A Mc and B stock solution was freshly prepared and combined in the well just before incubation. The proposed dilutions allowed us to observe the behavior of the mixture when the concentration of the main compound (Mc) was higher than that of the other isolated compounds (B) when they were at the same concentration and when the Mc was present in a lower concentration than B.

The egg hatching inhibition and lethal effect ratio were calculated by:Egg hatching inhibition ratio = OEHI/EOEHI(1)
where:OEHI: Observed egg hatching inhibition (%);EOEHI: Expected observed egg hatching inhibition (%).

An egg hatching inhibition ratio close to 1 indicated additivity, <1 subadditivity, and >1 synergism [11]
Lethal effect ratio = OEHI/EOEHI(2)
where:OLE: Observed lethal effect (%),ELE: Expected lethal effect (%).

A lethal effect ratio close to 1 indicated additivity, <1 subadditivity, and >1 synergism [11].

### 2.9. Statistical Analyses

Differences among lethality and egg hatching inhibition percentage means were compared using the Duncan test (*p* < 0.05). Lethal concentrations (LC_25_, LC_50_, and LC_90_) were calculated utilizing the PROBIT method integrated into the SAS statistical package 9

## 3. Results

### 3.1. Isolation and Identification of Compounds

Compounds **1** and **2** (C1F3: 1-nonacosanol/hentriacontane) ^13^DEPTQ (500 MHz, D_6_H_6_) δ (ppm) (Appendix A): δ 64.66 (-RCH_2_-OH), δ 34.87 (CH_2_), δ 29.51 (CH_2_), δ 23.47 (CH_2_), and δ 14.72 (CH_3_). ^1^H-NMR (500 MHz, D_6_H_6_) δc (ppm) (Appendix A): δ 4.08 (t, *J* = 6.7 Hz, 1H), δ 2.21 (t, *J* = 7.4 Hz, 1H), δ 2.08 (t, *J* = 7.4 Hz, 1H), δ 1.36–1.23 (m), and δ 0.93–0.89 (m). The signals at δ 64.66 and 4.08 (t, *J* = 6.7 Hz, 1H) provide evidence of a primary alcohol. Additionally, signals at δ 34.87, δ 29.51, and δ 1.36–1.23 (m) correspond to R-CH_2_-R units of saturated alkanes. The signals at δ 23.47 and δ 2.21 (t, *J* = 7.4 Hz, 1H) and 2.08 (t, *J* = 7.4 Hz, 1H) indicate CH_2_ groups bound to a CH_3_ moiety. Finally, the signals at δ 14.72 and δ 0.93–0.89 (m) correspond to a CH_3_ group. Aromatic or unsaturated carbons are not detected in accordance with the chemical shifts and the absence of UV absorption. Furthermore, the multiplicity revealed in DEPQ indicates the absence of any CH groups. Based on these findings, compounds **1** and **2** are linear saturated organic compounds featuring a terminal OH group.

Compound **1** showed a molecular ion of [M + H]^+^ = 425 *m*/*z* (Appendix A) and compound **2** ([M + H]^+^ = 437) (Appendix A). 1-Nonacosanol (**1**) and hentriacontane (**2**) (Figure 1) match the previously mentioned chemical and structural characteristics and also exhibit the expected molecular ion. According to GC-MS, C1F3 is composed of 31.343% 1-noncacosanol and 52.612% hentriacontane (Appendix A).

Compound **3** (Cinic acid): C1F5 HPLC-DAD retention time (15.749 min), UV spectra (λmax 211.0 nm) (Appendix A), and mass spectra [M + H]^+^ = 283 *m*/*z* (Appendix A) coincide with the cinic acid (Figure 2) previously reported ([7]).

Compound 4 (Peruvin): (^1^H-NMR and DEPTQ 500 MHz, CD_3_Cl_3_, δ (ppm)) (Appendix A): C1F6 ^13^CNMR data show 15 signals characteristic of sesquiterpene (Table 2). The δ 216.14 signal (C-4) corresponds to a ketone carbonyl and δ 168.20 to an ester or carboxylic acid carbonyl. The δ 141.33 (C-11) and δ 122.72 (C-13) signals were alkene types, and the aromatic signal was discarded due to the absence of signals between δ 7.00 and δ 8.00 in the ^1^H -NMR spectrum. δ 83.86 and 80.84 correspond to C-O signals. Methyl groups were observed at δ 22.82 (C-14) and δ 18.91 (C-15).

HMBC analysis shows a correlation between C-12 and H-8 (δ 4.95 ddd, *J* = 12.0, 8.0, 3.1 Hz) indicating the presence of a lactone group (Appendix A). The occurrence of signals corresponding to ketone, ester carbonyl, and alkene is commonly observed in guaianolides and pseudoguaianolides (5-7 bicyclic compounds) [12]. The presence of a methyl group at the C-5 ring junction (C-14 δ 22.82) and C-10 (C-15 δ 18.91) confirms the presence of a pseudoguaianolide.

According to spectroscopic data and mass spectra ([M + H]^+^ = 265 *m*/*z*) (Appendix A), compound **3** can be peruvin (Figure 3). The C1F3 NMR data were compared with peruvin [13], confirming the presence of the sesquiterpene lactone.

### 3.2. Lethal Effect against Infective Larvae L_3_ and Egg Hatching Inhibition of H. contortus from Compounds Isolated in the Ethyl Acetate Extract of A. cina

The isolated compounds were tested against *H. contortus* L_3_ and eggs. All compounds exhibited lethal effects (LEs) against infective larvae and demonstrated more significant activity than the ethyl acetate extract (EAE) except for 1-nonacosanol/hentriacontane, which had no difference with EAE at LC_90_. Based on LC_90_ and LC_50_, cinic acid showed the highest lethal effect, followed by peruvin and 1-nonacosanol/hentriacontane (Table 1).

On the other hand, cinic acid showed no inhibition of larvae hatching in the egg-hatching inhibition (EHI) assay at the tested concentrations. Higher concentrations of cinic acids were not evaluated to obtain an effect because the highest concentration evaluated was 2 mg/mL, which is close to the ethyl acetate (EAE) LC_50_ (2.422 (2.287–2.555)) with no egg hatching inhibition (EHI) activity. This means that cinic acid is not responsible for EAE activity related to EHI, and in a practical scenario, it is better to use EAE than cinic acid. Also, a large amount of pure compound would be necessary to perform the experiment.

Compared to EAE, 1-nonacosanol/hentriacontane and peruvin showed better EHI, with 1-nonacosanol/hentriacontane presenting the most significant EHI activity, followed by peruvin.

Peruvin was chosen as the main compound (Mc) for leading binary mixtures based on its EHI, LE, and yield percentage in the EAE.

### 3.3. Pharmacodynamic Interactions

Binary mixtures were prepared to investigate pharmacodynamic interactions. Ratios close to 1 indicate additivity, ratios < 1 suggest subadditivity, and ratios > 1 suggest synergism [14]. For EHI, only mixtures of peruvin-1-nonacosanol/hentriacontane (Table 3) were evaluated due to the absence of EHI in cinic acid.

All proportions of peruvin/1-nonacosanol/hentriacontane exhibited a ratio close to 1, indicating additivity. On the other hand, binary mixtures evaluating LE showed synergism in 0.50LC_25_/0.50LC_25_ and 0.25LC_25_/0.75LC_25_ peruvin-1-nonacosanol/hentriacontane, with ratios of 4.87 and 1.75, respectively (see Table 4).

Peruvin–cinic acid mixtures also demonstrated synergism at 0.50LC_25_/0.50LC_25_ with a ratio of 1.42. Binary mixtures with peruvin and the Mc allowed us to find pharmacodynamic interactions.

## 4. Discussion

A plant extract is a complex matrix of various plant secondary metabolites, making it challenging to determine the specific compound responsible for pharmacological activity [15]. While bio-guided separation is effective in identifying the active compound in a plant extract, it has limitations; notably, it does not consider potential interactions (pharmacological interactions) between compounds of different polarities. Therefore, in the present study, binary mixtures are proposed as simpler matrices than a plant extract to comprehend better the possible mode of action of the plant extracts, the EAE of *A. cina* in this case. To perform binary mixtures, it is necessary to set a main compound (Mc) and a compound of interest (Ci). The established Mc is desirable to be a significant component of the plant extract and well-documented regarding its biological evaluation. The CIs are preferable to be at least one type of secondary metabolite present in the plant extract from which they are isolated. Unfortunately, there is a lack of information about the chemical profile of *A. cina*, and the only reported compound of EAE is sesquiterpene cinic acid [7]. But, this work provides information of *A. cina* and its EAE chemical profile, describing the presence of sesquiterpene lactones, such as peruvin, fatty alcohol like 1-nonacosanol, and alkane such as hentriacontane.

The isolated compounds were tested against *H. contortus* eggs and larvae L_3_, demonstrating anthelmintic activity. The 1-nonacosanol/hentriacontane mixture exhibited the most significant inhibition of larvae hatching in the egg hatching inhibition (EHI) assay and displayed a lethal effect (LE) on LC_50_ superior to EAE LC_50_. The anthelmintic activity observed could be attributed to its potential for an interaction with cellular membrane phospholipids, leading to alterations in the membrane fluidity and functionality of both eggs and larvae [16]. Also, the inhibition of larvae hatching can increase with the number of carbons [17]. The large difference between LC_50_ and LC_90_ in LE can be attributed to a flat dose–response curve. This indicates that increases in dosage do not result in a proportional increase in lethal effect. A steeper curve would be expected in a highly potent drug, where a slight increase in dosage would produce a large increase in effect. Additionally, the affinity and efficacy of nonacosanol/hentriacontane for the biological target may reflect a low or moderate affinity, requiring higher concentrations to achieve a lethal effect in most of the population due to the saturation of action sites or resistance mechanisms that limit the efficacy [18].

In the cases of peruvin and cinic acid, these compounds exhibit a significant structural similarity. Both are sesquiterpenes, but the differences lie in the absence of the γ-lactone ring closed at C8 and the presence of an OH group at C8 in cinic acid compared to peruvin. Peruvin displayed EHI and LE activities, whereas cinic acid did not show an inhibition of larvae hatching. The effectiveness of peruvin can be attributed to the existence of the α-methylene, γ-lactone system. This system acts as a Michael acceptor, facilitating an interaction with thiol groups found in proteins [19]. The presence of the α-methylene, γ-lactone system is essential for the EHI activity of sesquiterpene lactones. On the other hand, those structural changes between peruvin and cinic acid enhance LE. Cinic acid displayed a 1.85-fold increase at LC_90_ and a 7.11-fold increase at LC_50_ compared to peruvin. It suggests that the α-methylene, γ-carbonyl system is responsible for LE. Peruvin was selected as the Mc for binary mixtures based on the results obtained for EHI, LE, yield percentage, and purity. There was no previous report on anthelmintic activity against *H. contortus* for peruvin. 

The methodological approach involving binary mixtures with Mc and Ci identified pharmacological interactions, as the interaction presented by the peruvin-1-nonacosanol/hentriacontane mixture exhibited synergism in LE and additivity in EHI. The synergism in LE may be attributed to the collaborative action of peruvin and 1-nonacosanol/hentriacontane, which operate through different mechanisms of action when working together. Peruvin, resembling ivermectin as a lactone and with a strong affinity for thiol groups in proteins [20], may act similarly to ivermectin by selectively binding to glutamate-gated chloride ion channels present in the muscle and nerve cells of *H. contortus*. This binding enhances the permeability of the cell membrane to chloride ions, resulting in cell hyperpolarization, paralysis, and eventual parasite death [21]. Additionally, 1-nonacosanol/hentriacontane may influence cellular membrane fluidity and functionality [17]. It is potentially amplifying permeability to chloride ions and LE up to 4.87-fold. This binary mixture can also be tested in in vivo models due to the low dose required: 0.035 mg/kg of peruvin combined with 0.0017 mg/kg of hentriacontane.

The peruvin–cinic acid mixture also presents synergism in LE, and it is hypothesized that the synergism (1.42-fold) is attributed to the collaborative mechanism of action of the α-methylene, γ-lactone system of peruvin and the α-methylene, γ-carbonyl system of cinic acid. 

Additionally, these results confirm that plants from the Asteraceae family and sesquiterpene lactones can be an alternative against *Haemonchus contortus* [22,23,24,25].

## 5. Conclusions

This is an initial exploratory study identifying bioactive compounds isolated from the ethyl acetate extract of *Artemisia cina*, establishing its in vitro anthelmintic activity and its LC_50-90_. LC_90_ ensures that therapeutic doses in in vivo experiments are below the lethal level, minimizing animal suffering as much as possible. This information is indispensable for conducting in vivo studies, as it would be unethical to use experimental animals without first demonstrating the in vitro activity and the doses to be administered. The next step would be to demonstrate anthelmintic activity in a murine model and sheep. In vivo experiments are necessary to determine the real effectiveness of the ethyl acetate extract, isolated compounds, and the synergistic binary mixture (Peruvin–hentriacontane/1-nonacosanol), given that the internal conditions of the animal (pH and enzymes) could degrade the bioactive compounds, potentially reducing or increasing their activity.

The methodological approach employing binary mixtures with a main compound and the compounds of interest represents a strategic avenue for uncovering valuable pharmacological interactions. The study of pharmacodynamic interactions via binary mixtures utilizing a main compound in conjunction with compounds of interest isolated from the same extract significantly enhances our comprehension of the mechanism of action of *A. cina* ethyl acetate extract. These interactions are crucial for the formulation of an *A. cina* pharmaceutical preparation with anthelmintic activity against *H. contortus*.

## Figures and Tables

**Figure 1 pathogens-13-00509-f001:**
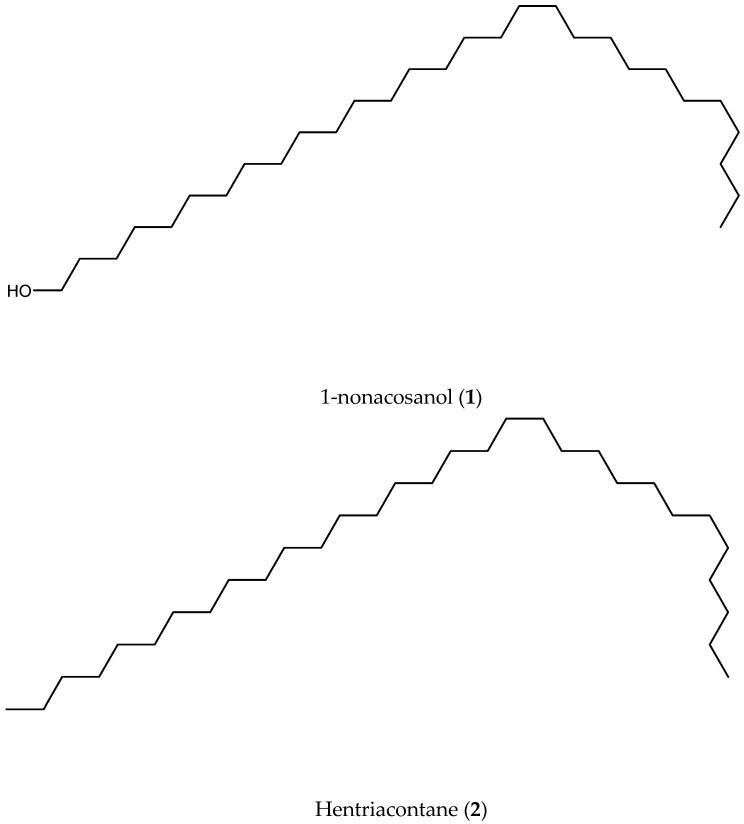
Compounds identified from C1F3 present in *Artemisia cina* ethyl acetate extract.

**Figure 2 pathogens-13-00509-f002:**
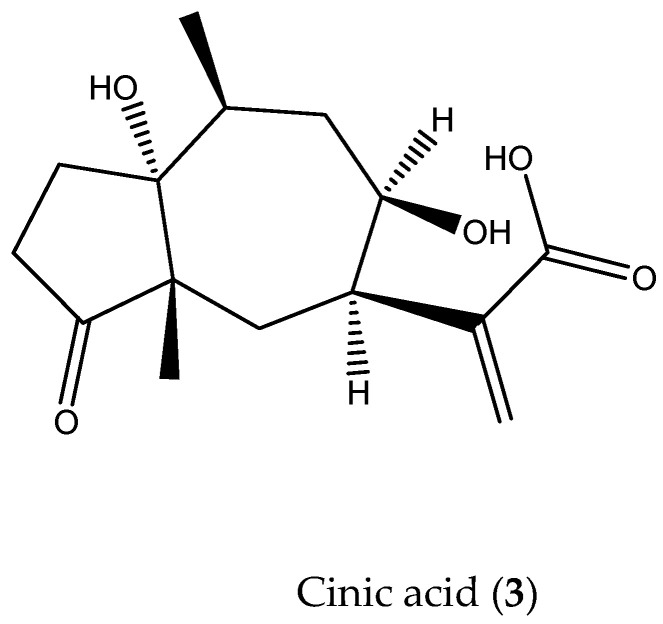
Cinic acid isolated from *Artemisia cina* ethyl acetate extract.

**Figure 3 pathogens-13-00509-f003:**
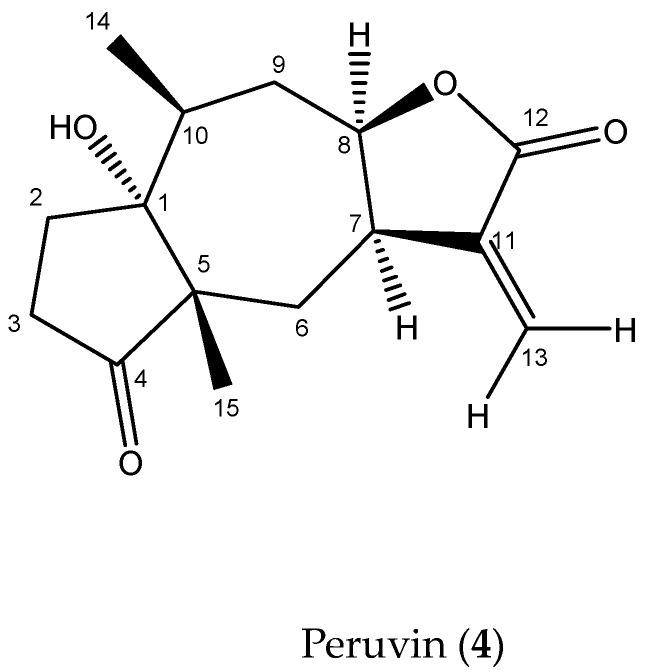
Peruvin isolated from *Artemisia cina* ethyl acetate extract.

**Table 1 pathogens-13-00509-t001:** Egg hatching inhibition and larvicidal effect of *A. cina* isolated compounds from the ethyl acetate extract against eggs and infective larvae of *H. contortus*.

Treatments	LC_90_ (mg/mL)	LC_50_ (mg/mL)	LC_25_ (mg/mL)
Egg hatching inhibition			
EAE	3.849 (3.640–4.107) ^b^	2.422 (2.287–2.555) ^c^	1.897 (1.739–2.037) ^c^
C1F3	1.322 (1.214–1.454) ^a^	0.459 (0.423–0.496) ^a^	0.263 (0.235–0.292) ^a^
Cinic acid	No effect	No effect	No effect
Peruvin	1.582 (1.402–1.687) ^a^	1.095 (1.055–1.135) ^b^	0.902 (0.855–0.943) ^b^
Larvicidal effect			
EAE	3.302 (3.265–3.569) ^C^	2.564 (2.457–2.654) ^C^	1.786 (1.622–1.928) ^C^
C1F3	2.607 (1.810–4.432) ^C^	0.060 (0.034–0.089) ^A^	0.003 (0.002–0.003) ^A^
Cinic acid	0.220 (0.192–0.254) ^A^	0.018 (0.015–0.021) ^A^	0.005 (0.003–0.006) ^A^
Peruvin	0.410 (0.377–0.466) ^B^	0.128 (0.119–0.137) ^B^	0.069 (0.063–0.076) ^B^

C1F3: 1-nonacosanol/hentriacontane. Same letters indicate no significant difference between groups. Duncan α < 0.05.

**Table 2 pathogens-13-00509-t002:** ^1^H -NMR and DEPTq spectroscopy data of compound **1** (CD_3_COCD_3_, 500 MHz).

Position	DEPTq	^1^H (*J* in Hz)	^13^C (ppm)
1	C	-	83.86
2	CH_2_	α 2.44 (1H, m)	32.76
		β 1.72 (1H, m)	
3	CH_2_	α 2.44 (1H, m)	31.67
		2.44 1H, (m)	
4	C	-	216.14
5	C	-	54.99
6	CH_2_	α 2.55 (1H, dd, *J* = 15.1, 5.2 Hz)	35.71
		β 1.47 (1H, dd, *J* = 15.1, 13.0 Hz)	
7	CH	3.83 (1H, m)	38.98
8	CH	4.95 (1H, ddd, *J* = 12.0, 8.0, 3.1 Hz)	80.84
9	CH_2_	α 2.19 (1H, ddd, *J* = 13.7, 5.0, 3.1 Hz)	36.78
		β 1.85 (1H, dt, *J* = 13.7, 12.0 Hz)	
10	CH	2.06 (1H, ddq, J = 12.2, 4.9, 7.3)	41.25
11	C	-	141.33
12	C	-	170.61
13	CH_2_	Ha 6.23 (1H, d, *J* = 2.8),	122.72
		Hb 5.64 (1H, d, *J* = 2.4 Hz)	
14	CH_3_	1.18 (3H, d, *J* =7.3)	22.82
15	CH_3_	1.06 (3H, s)	18.91

**Table 3 pathogens-13-00509-t003:** Fixed proportions scheme of peruvin-1-nonacosanol/hentriacontane binary mixture on *H. contortus* egg hatching inhibition activity.

Binary Mixture	Expected *H. contortus* EHI (%)	Observed *H. contortus* EHI (%)	Ratio
Water		2.1 ± 1.92 ^a^	
Ivermectin		100 ± 0.00 ^c^	
P-C1F3 proportion			
1.00/0:00	25.00	19.94 ± 6.38 ^b^	
0.75/0.25	22.45	23.24 ± 5.51 ^b^	1.03
0.50/0.50	24.96	24.98 ± 4.23 ^b^	1.00
0.25/0.75	27.47	25.60 ± 5.82 ^b^	0.93
0.00/1.00	25.00	29.99 ± 4.70 ^b^	

P-C1F3: peruvin-1-nonacosanol/hentriacontane. EHI: egg hatching inhibition. Same letters indicate no significant difference between groups. Duncan α < 0.05. Inhibition ratio close to 1 indicates additivity, <1—subadditivity, and >1—synergism. Incubated for 48 h.

**Table 4 pathogens-13-00509-t004:** Fixed proportions scheme of peruvin-1-nonacosanol/hentriacontane and peruvin–cinic acid binary mixtures on the lethal effect of *H. contortus* L_3_.

Binary Mixture	Expected LE on *H. contortus* L_3_ (%)	Observed LE on *H. contortus* L_3_ (%)	Ratio
Water		3.53 ± 0.91 ^a^	
Ivermectin		100 ± 0.00 ^f^	
P-C1F3 proportion			
1.00/0:00	25.00	21.36 ± 4.18 ^b^	
0.75/0.25	16.56	19.78 ± 3.19 ^b^	1.19
0.50/0.50	19.96	97.33 ± 2.22 ^e^	4.87
0.25/0.75	14.97	26.27 ± 5.64 ^bc^	1.75
0.00/1.00	25.00	18.54 ± 8.0 ^b^	
P-C proportion			
1.00/0:00	25.00	23.28 ± 4.60 ^b^	
0.75/0.25	25.85	22.88 ± 6.18 ^b^	0.99
0.50/0.50	28.39	40.42 ± 8.25 ^d^	1.42
0.25/0.75	30.95	25.60 ± 5.82 ^b^	0.85
0.00/1.00	25.00	33.51 ± 4.96 ^cd^	

P-C1F3: peruvin-1-nonacosanol/hentriacontane, P-C: peruvin–cinic acid, and LE: lethal effect. The same letters indicate no significant difference between groups. Duncan α < 0.05. Inhibition ratio close to 1 indicates additivity, <1—subadditivity, and >1—synergism. Incubated for 24 h.

## Data Availability

The data presented in this study are available within the article or Appendix A.

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
