# Peer review of "Understanding Artemisia cina Ethyl Acetate Extract’s Anthelmintic Effect on Haemonchus contortus Eggs and L3 Larvae: The Synergism of Peruvin Binary Mixtures"

_pathogens, 2024, doi:10.3390/pathogens13060509_

Round 1
Reviewer 1 Report
Comments and Suggestions for Authors
The paper “Understanding Artemisia cina Ethyl Acetate Extract Anthelmintic Effect on Haemonchus contortus Eggs and L3 Larvae: Synergism of Peruvin binary mixtures” presents unprecedented and interesting results for the area, as there are few published works that deal with pharmodynamic interactions. However, it is necessary to improve the discussion of results.
Corrections were made in the attached file. Corrected file follows.

Author Response
Dear reviewer 1
We wanted to extend our sincere gratitude for taking the time to review our manuscript and provide detailed feedback. Your insights and suggestions were invaluable in improving the quality and clarity of our work.
The changes were made. Please see the attachment.

Reviewer 2 Report
Comments and Suggestions for Authors
This paper investigates the effect of ethyl acetate extract and identified compounds isolated from Artemisia cina on egg hatching and L3 larvae of Haemonchus contortus. I have the following comments:
Provide a list of abbreviations in alphabetical order after keywords.
lines 150/3: since authors tested compounds at different concentrations, suggest they add the graphs plus controls in the results section.
line 154: did all the eggs hatch in distilled water?
line 246: why give docosanol here and line 248 but nonacosanol in tables and elsewhere in text?
Table 2: in control experiments did any L3 larvae de over 24 hours? Comment on number of eggs that hatched in control experiments. Under cinic acid in table, put no effect rather than -, the latter suggests it was not tested. Please comment on large difference in effect between LC90 and LC50 for nonacosanol/hentriacontane mixture.
line 253: did authors try higher concentrations of cinic acid to try and obtain an effect? Note: cinic not clinic.
line 261: suggest move Table S1 and put as a third column in Table 2.
Table 3: add to legend, incubated for 28 hours. Did all the eggs hatch in water? Does P-M stand for peruvin-mixture? Suggest authors put ivermectin above P-M proportion in table. Perhaps add water and indicate 0. line 268/9: there are no letters given in Table 3.
Table 4 legend: add incubated for 24 hours. As above, invert ivermectin and P-M and P-C proportion and add water 0.
Minor points: line 247: cinic, not clinic and should it be lethal rather than legal? line 310: its. line 311: potential for the interaction. line 318: acts.
Comments on the Quality of English Language
The English is very good but helpful if an editor checks for any minor errors.
Author Response
Dear Reviewer 2,
We wish to express our genuine appreciation for dedicating your time to review our manuscript and offering insightful feedback. Your suggestions have been immensely helpful in enhancing the quality and clarity of our work.
We have incorporated the suggested changes, and we invite you see the attachment

Reviewer 3 Report
Comments and Suggestions for Authors
Dear authors,
the topic of this manuscript is of greatest interest to the readers, and hopefully, the results will bring new possibilities to managing H. contortus infections. The article is clear and well written in my opinion.
Comments on the Quality of English Language
Not being a native speaker myself, I find the manuscript to be well written and easily readable.
Author Response
Dear reviewer 3
We wanted to extend our sincere gratitude for taking the time to review our manuscript and your comment.

Reviewer 4 Report
Comments and Suggestions for Authors
- The title of the manuscript could be more direct. For example, what is the need for the term "understanding"?
Abstract
- In this section, tell us which proportion of the compound mixtures was the most effective?
M&M, results and discussion
-The title of topic 2.7 does not provide for ovicidal test.
- Inform which aspects are evaluated in the eggs and larvae after incubation in the ovicidal and larvicidal test.
- Although it is known that larvae hatch within a period of 24 h, many studies use a period of 48 hours to verify the effect of compounds in ovicidal tests. Why did the authors prefer to use only 24 h?
- Inform about the practical significance of the LC90 results.
- Throughout the manuscript the authors talk about the egg hatching, but the larva is the one that hatches. To review.
- Report on the yield of these compounds. What would be the best method of obtaining it on a large scale, isolation from plants or chemical synthesis? Would the amount of compounds be feasible to be used in in vivo tests?
- Inform whether the mixtures were carried out in the well itself or at some point before incubation. Inform the reader about the stability of the mixture. Inform about criteria relating to the dilution of substances in the medium used.
- Discuss the importance of translating these results to in vivo tests. We know that anthelmintics need to have an effect on adults. The egg and L3 stages occur in the environment, so these tests are for screening purposes only. The in vitro test on adults would have been interesting to approximate the results of in vivo tests on the target species.
Comments on the Quality of English Language
The spelling of the manuscript needs to be checked, especially the summary.
Author Response
Dear Reviewer 4,
We wish to express our genuine appreciation for dedicating your time to review our manuscript and offering insightful feedback. Your suggestions have been immensely helpful in enhancing the quality and clarity of our work.
We have incorporated the suggested changes, and we invite you to review the attachment
